# A Quantitative Proteomics Approach to Gain Insight into NRF2-KEAP1 Skeletal Muscle System and Its Cysteine Redox Regulation

**DOI:** 10.3390/genes12111655

**Published:** 2021-10-21

**Authors:** Rafay Abu, Li Yu, Ashok Kumar, Lie Gao, Vikas Kumar

**Affiliations:** 1Mass Spectrometry and Proteomics Core Facility, University of Nebraska Medical Center, Omaha, NE 68198, USA; rafay.abu@unmc.edu; 2Department of Cellular and Integrative Physiology, University of Nebraska Medical Center, Omaha, NE 68198, USA; liyu@gmail.com; 3Centre for Systems Biology and Bioinformatics (U.I.E.A.S.T), Panjab University, Chandigarh 160014, India; ashokbiotech@gmail.com; 4Department of Genetics, Cell Biology and Anatomy, University of Nebraska Medical Center, Omaha, NE 68198, USA

**Keywords:** Nrf2/Keap1 system, skeletal muscle, sedox cysteine, quantitative proteomics, IPA

## Abstract

Mammalian skeletal muscle (SkM) tissue engages the Nrf2-Keap1-dependent antioxidant defense mechanism to respond adaptively to stress. Redox homeostasis mediated by the reversible modification of selective cysteines is the prevalent mode of regulation. The protein targets of SkM redox regulation are largely unknown. We previously reported the proteomic profiles of soleus (Sol) and extensor digitorum longus (EDL) with Nrf2 or Keap1 gene deletion, using SkM-specific Nrf2 or Keap1 knockout models; iMS-Nrf2flox/flox; and iMS-Keap1flox/flox. Here, we employed these two animal models to understand the global expression profile of red tibialis anterior (RTA) using a label free approach and its redox proteomics using iodoacetyl tandem mass tag (iodoTMT^TM^)-labeled cysteine quantitation. We quantified 298 proteins that were significantly altered globally in the RTA with Nrf2 deficiency but only 21 proteins in the Keap1 KO samples. These proteins are involved in four intracellular signaling pathways: sirtuin signaling, Nrf2 mediated oxidative stress response, oxidative phosphorylation, and mitochondrion dysfunction. Moreover, we identified and quantified the cysteine redox peptides of 34 proteins, which are associated with mitochondrial oxidative phosphorylation, energy metabolism, and extracellular matrix. Our findings suggest that Nrf2-deficient RTA is implicated in metabolic myopathy, mitochondrial disorders, and motor dysfunction, possibly due to an enhanced oxidative modification of the structure and functional proteins in skeletal myocytes.

## 1. Introduction

Skeletal muscle in mammals is tightly regulated under stress conditions and dysfunctional physiological processes. The Nrf2-Keap1 system is one of the most efficient antioxidant defenses that is active in muscle tissues in response to reactive oxygen species (ROS) and related species that have a damaging effect on skeletal myocytes.

Nrf2 is a master transcription factor that is responsible for cellular redox homeostasis. Under baseline conditions, Nrf2 is sequestered in the cytosol by Keap1 and is degraded through the ubiquitin proteasome system (UPS). In oxidative stress, when reactive oxygen species (ROS) increases, Nrf2 escapes from Keap1 and translocates to the nucleus, upregulating a large group of antioxidant-associated proteins to restore redox balance. It has been well-recognized that Nrf2 plays a critical role in protecting skeletal muscle (SkM) function, particularly during exercise, when contracting SkM generates a large amount of reactive oxygen species (ROS) [1]. In addition, impaired Nrf2 signaling contributes to aging-related SkM dysfunction [2], whereas the Nrf2 activator, sulforaphane, prevents this pathology [3]. Employing proteomics and bioinformatics analyses of SkM with Nrf2- or Keap1- deficiency, we identified over 200 proteins governed by Nrf2 that are involved in antioxidant defense, mitochondrial function, oxidoreduction coenzyme metabolism, cellular detoxification, and many other key biology processes [4]. Interestingly, in this study, we noticed that Nrf2 deletion-induced alteration of protein expression was predominantly observed in the extensor digitorum longus (EDL), a fast-switch muscle composed of glycolytic type IIB fibers. In contrast, Keap1 deletion-induced differential expression of proteins was mostly found in the soleus (Sol), a slow-switch muscle composed of oxidative Type I and Type IIA fibers, suggesting a fiber type-specific significance of the Nrf2/Keap1 system in SkM.

Post-translational modification of cysteine is a key regulatory event that modulates the protein function, its localization, and interaction with potential partners [5,6]. Cysteine thiols are inherently sensitive to ROS and related species and can be covalently modified either reversibly or irreversibly. The redox modification of protein cysteines is implicated in several tissue-specific regulatory processes [7]. A disturbed physiological redox state or the dysregulation of redox signaling is one of the most recognized underlying causes of physiological conditions, including tissue aging, metabolic disorder, mitochondrion, motor dysfunction, and neurodegenerative disorders [8,9].

In the present study, we investigated the effects of Nrf2- or Keap1-deletion on the proteomic profile of red tibialis anterior (RTA), a muscle composed of Type IIX. In addition, the proteins with oxidized cysteine were identified in these samples to determine the influence of redox statutes on protein function when Nrf2 or Keap1 were deleted.

## 2. Materials and Methods

### 2.1. Inducible Skeletal Muscle-Specific Nrf2 and Keap1 Knockouts

All animal procedures were conducted in accordance with the guidelines of the National Institutes of Health Guide for the Care and Use of Laboratory Animals and conformed to ARRIVE Guidelines (https://www.nc3rs.org.uk/arrive-guidelines, accessed on 21 June 2019), as approved by the Animal Care and Use Committee of the University of Nebraska Medical Centre (UNMC-IACUC Protocol #18-174-02). The iMS-Nrf2flox/flox and iMS-Keap1flox/fox mice were produced in our laboratory by crossing the HSA-MCM line with Nrf2flox/flox and Keap1flox/flox lines and was validated by genotyping, RT-PCR, and Western blot, as reported in previous publications [4]. Sixteen male mice, eight iMS-Nrf2flox/flox, and eight iMS-Keap1flox/fox, were used in the present experiment, with four mice acting as a wildtype control and four mice acting as gene knockout samples in each group. The gene inactivation was induced by intraperitoneal injections of tamoxifen (2 mg/0.2 mL day^−1^, Sigma-Aldrich, St. Louis, MO, USA; Cat. No. T5648) for five consecutive days. Controls were vehicle (Veh, 15% ethanol in sunflower seed oil, 0.2 mL day^−1^ for 5 days)-treated mice. Five months post tamoxifen or vehicle treatment, the mice were euthanized by the administration of 5% isoflurane breathing had been stopped for 1 min. The mice were decapitated, and bilateral RTA were removed from the hindlimb and stored at −80 °C for analysis.

### 2.2. Mass Spectrometry-Based Proteomics

Mice were sacrificed by means of CO_2_ inhalation. The muscle tissues were collected and snap-frozen in liquid nitrogen. Samples were homogenized in RIPA buffer (50 mM Tris-HCl, 195 mM NaCl, 2 mM EDTA, 1% NP-40, 0.1% SDS) with 1% protease inhibitor cocktail (Abcam, ab65621) and 100 mM NEM, and the protein was extracted by centrifugation at 20,000× *g* at 4 °C for 20 min. Protein concentration was quantified by protein assay (Pierce; Rockford, IL, USA). For global proteomics, 100 µg of protein per sample from four biological replicates per group was reduced and alkylated with 10 mM DTT at 55 °C and 50 mM iodoacetamide at RT, respectively. For redox proteomics, 250 µg of protein from four biological replicates per group in RIPA buffer and 100 mM NEM was buffer exchanged with 100 mM ammonium bicarbonate to remove excess NEM using Zeba desalting columns with a 7 KDa cut-off (Thermo Scientific™; Waltham, MA, USA). Free thiols were labeled using 0.8 mg Cys thiol-reactive iodoacetyl tandem mass tag (iodoTMT^TM^) sixplex reagent (Pierce; Rockford, IL, USA) according to manufacturer protocols (Thermo Scientific ™). Detergent was removed by chloroform/methanol extraction. The protein pellet was re-suspended in 50 mM ammonium bicarbonate and was digested with MS-grade trypsin (Pierce; Rockford, IL, USA) overnight at 37 °C. Peptides cleaned with PepClean C18 spin columns (Thermo Scientific™; Waltham, MA, USA) were re-suspended in 2% acetonitrile (ACN) and 0.1% formic acid (FA). IodoTMT-labeled peptides were further enriched using anti-TMT antibody resin from Thermo Scientific™. An amount of 500 ng of each sample was loaded onto the trap column Acclaim^TM^ PepMap^TM^ 100 C18 75 µm × 2 cm LC Columns (Thermo Scientific™; Waltham, MA, USA) at a flow rate of 4 µL/min and were then separated with a Thermo RSLC Ultimate 3000 (Thermo Scientific™; Waltham, MA, USA) on a Thermo Easy-Spray^TM^ PepMap RSLC C18 2 µm, 100 A°, 75 µm × 50 cm column (Thermo Scientific™; Waltham, MA, USA) with a step gradient of 4–25% solvent B (0.1% FA in 80% ACN) from 10–130 min and of 25–45% solvent B for 130–145 min at 300 nL/min and 50 °C, resulting in a 180 min total run time. Eluted peptides were analyzed by a Thermo Orbitrap Fusion Lumos Tribrid (Thermo Scientific™; Waltham, MA, USA) mass spectrometer in a data-dependent acquisition mode. A survey full scan MS (from *m*/*z* 350–1800) was acquired in the Orbitrap with a resolution of 120,000. The AGC target for MS1 was set as 4 × 10^5^, and the ion filling time was set as 100 ms. The most intense ions with charge states 2–6 were isolated in a 3 s cycle and were fragmented using HCD fragmentation with 35% normalized collision energy and were detected at a mass resolution of 30,000 at 200 *m*/*z*. The AGC target for MS/MS was set as 5 × 10^4^, and the ion filling time was set to 60 ms, and dynamic exclusion was set for 30 s with a 10 ppm mass window. Protein identification was performed by searching MS/MS data against the swiss-prot Mus musculus protein database downloaded on 16 February 2021 using the in-house mascot 2.6.2 (Matrix Science, Boston, MA, USA) search engine. The search was set up for full tryptic peptides with a maximum of two missed cleavage sites. The acetylation of protein N-terminus and oxidized methionine were included as variable modifications, and carbamidomethylation and addition of iodoTMT sixplex on cysteine thiols were set as a fixed modification for label-free and redox proteomics, respectively. The precursor mass tolerance threshold was set at 10 ppm, and the maximum fragment mass error was 0.02 Da. The significance threshold of the ion score was calculated based on a false discovery rate of ≤1%. Quantitative analysis was performed using progenesis QI proteomics 4.1 (Nonlinear Dynamics, Milford, MA, USA) for global expression and built-in iodoTMT workflow in Proteome Discoverer 2.4 (Thermo Scientific™; Waltham, MA, USA) for redox proteomics. Protein expression fold changes between Nrf2-KO/Nrf2-WT and Keap1-KO/Keap1-WT were represented as Log_2_FC^Nrf2-KO^ and Log_2_FC^Keap1-KO^, respectively.

### 2.3. Western Blot Analyses

Skeletal muscle tissues were homogenized in RIPA buffer (50 mM Tris-HCl pH7.4, 150 mM NaCl, 2 mM EDTA, 1% NP-40, 0.1% SDS) with 1% protease inhibitor cocktail (Abcam, ab65621), from which total protein was extracted by centrifugation at 20,000× *g*. The protein concentration of the extract was measured using a protein assay kit (Pierce; Rockford, IL, USA) and was then adjusted to equal volume in all samples with 2 × 4% SDS sample buffer. The samples were boiled for 5 min and were then loaded on a 7.5% SDS-PAGE gel (30 µg protein/10 µL per well) followed by electrophoresis using a Bio-Rad mini gel apparatus set at 40 mA/gel for 45 min. The fractionated protein on the gel was electrically transferred onto a polyvinyl difluoride membrane (Millipore). The membrane was first probed with the primary antibodies to Sdhd (ab189945) and Txnrd1 (ab124954) purchased from Abcam followed by the secondary antibody (HRP Goat Anti-Rabbit IgG Antibody, HRP, Thermo-Fisher Scientific, Waltham, MA, USA). After three washes with TBST, the membrane was treated with enhanced chemiluminescence substrate (Pierce; Rockford, IL, USA) for 5 min. The blots on the membrane were visualized and analyzed using a UVP BioImaging System (EpiChemi II Darkroom). The final reported data were normalized by Ponceau S staining.

### 2.4. Differential Proteomic and Pathway Enrichment Analyses

Proteins identified by mass spectrometry were quantified to identify differentially expressed proteins between each experimental and control condition. ANOVA *p*-value and absolute fold changes were used to identify differentially expressed proteins between the wildtype and gene knockout mice. A protein was considered to be differentially expressed if the *p*-value was ≤0.05 and the absolute fold change was ≥1.5. Gene enrichment analysis of differentially regulated proteins to identify known functions, pathways, and networks affected was performed using ingenuity pathway analysis (IPA) (Ingenuity Systems; Mountain View, CA, USA).

### 2.5. Proteomic and Bioinformatic Analyses

For all of the comparisons, the ANOVA *p* value was used. The cut off used for the differential expression analysis summary was *p* ≤ 0.05, and for the absolute fold change, the cut off was >1.5. IPA pathway analysis was also performed on genes with the same cut-off. For the volcano plots, the cut-off used to add gene names to differentially expressed proteins were absolute Log2 Fold change >1 and Anova *p*-value ≤ 0.05.

Structural comparisons of experimentally identified cysteine redox peptides were performed using UCSF chimera 1.15 (https://www.rbvi.ucsf.edu/chimera, accessed on 13 July 2021). Homologous protein structures from the protein database PDB (www.rcsb.org, accessed on 9 July 2021) were downloaded and superimposed in Chimera. Potential redox sensitive cysteine residues determined as iodoTMT modification were located and highlighted. For sequence logo analysis, peptides with +/− 5 amino acids flanking redox cysteine were subjected to multiple sequence alignment using Clustal W (www.ebi.ac.uk, accessed on 9 July 2021), and the sequence logo was created using webLogo v 2.8.2 University of California, Berkeley (weblogo.berkeley.edu, accessed on 15 July 2021).

## 3. Results

### 3.1. Nrf2/Keap1 Knockout Models: Mass Spectrometry Based Global Expression Analysis

To understand the global protein expression changes in skeletal muscle specific Nrf2 or Keap1 knockout mouse models, we investigated skeletal muscles proteins in Nrf2-WT (Nrf2 wild type), Nrf2-KO (Nrf2 knockout), Keap1-WT (Keap1 wild type), and Keap1-KO (Keap1 knockout). LCMS-based label-free quantitative proteomics analysis of skeletal muscle tissue was conducted between Nrf2-WT/ KO and Keap1-WT/KO mice (four biological replicates for each group). We quantified 722 proteins (363 upregulated and 358 downregulated in KO) between the Nrf2-WT/KO mouse muscle tissue, of which 298 proteins (42%) showed significant differential expression at a *p*-value of 0.05 or less. (Appendix A). Similarly, a total of 955 proteins (271 upregulated and 683 downregulated in KO) were identified and quantified between the Keap1-KO and Keap1-WT models. Interestingly, only 21 proteins (2%) were found to be significantly differentially expressed between the Keap1-WT and Keap1-KO groups. The expression profile in Nrf2-deficient muscle proteins was altered equally, with half of the proteins being upregulated and the other half being downregulated. In contrast, most of the proteins altered in Keap1-deficient muscle were downregulated (refer to Appendix A). Top 20 significantly expressed proteins between Nrf2-WT/KO and Keap1-WT/KO are shown (Figure 1A). The volcano plots highlight the proteins (red and green) with significant fold changes and the expression profiles of unique and overlapped proteins that are differentially expressed between the wild type and gene knockout muscles (Nrf2-KO/WT and Keap-KO/WT). Pairwise comparisons of significant proteins with a minimum fold of 1.5 are represented between the KO and WT groups (Figure 1B).

We found that there were six common proteins that were significantly altered between the Nrf2 and Keap1-deficient muscle proteins (see Table 1). With the exception of cytoskeletal muscle filament protein Synm and Gbe1, a protein associated with glycogen metabolism, the most commonly altered proteins were mitochondrial proteins (Sdhd, Pcca) and proteins involved in redox homeostasis (Tsnrd1, Gstm2).

To validate the sensitivity and accuracy of mass spectrometry, we employed the Western blotting technique to evaluate the expression levels of Sdhd and Txnrd1, with two proteins selected from Table 1 acting as the representatives of porteins that were upregulated/downregulated by Nrf2-KO/Keap1-KO separately (Sdhd) or downregulated/upregulated by Nrf2-KO/Keap1-KO separately (Txnrd1). As it can be seen from the left panels in Figure 2C, the band density of Sdhd was significantly increased in Nrf2 KO muscles compared to its band density in Nrf2 WT (2.93 ± 0.27 vs. 2.08 ± 0.13, *n* = 4, *p* = 0.003), but it was decreased in Keap1 KO muscle compared to in Keap1 WT muscle (1.75 ± 0.25 vs. 2.46 ± 0.39, *n* = 4, *p* = 0.036). On the contrary, Txnrd1 expression displayed an opposite consequence to Sdhd when Nrf2 or Keap1 was deleted (right panels in Figure 2C). These changes are consistent with the mass spectrometry data shown in Table 1.

### 3.2. MS Quantitative Analysis of Redox Regulated Proteins

We further investigated the role of the Nrf2/Keap1 system in regulating the redox homeostasis of muscles. To determine the cysteine oxidation profiles, muscle proteins were harvested from Nrf2- and Keap1-deficient mice. Four biological replicates from each group, Nrf2-KO, Nrf2-WT, Keap1-KO, and Keap1-WT, were harvested for skeletal their muscle proteins and were subjected to iodoTMT reagent labeling for the quantification of reversibly oxidized thiols. Free thiols were blocked with NEM (see methods section). In contrast to global expression change, we observed a relatively lower number of significantly altered redox-sensitive peptides in Nrf2/Keap1 deletion knockouts. A total of 85 peptides were quantified as harboring iodoTMT modification at a reversibly oxidized cysteine residue in both Nrf2- and Keap1-deficient muscle. Furthermore, we identified 15 reversibly oxidized peptides in Nrf2-deficient (Table 2) and 19 reversibly oxidized peptides in Keap1-deficient muscle (Table 3) that were differentially expressed in the knockout mice with *p*-values < 0.05.

#### 3.2.1. Nrf2-KO/WT Reversibly Oxidized Redox Peptides

The differentially expressed redox peptides and their corresponding protein in Nrf2 knockout muscle reveal the complex interplay of the functional proteins regulated in response to oxidative stress. This analysis provides evidence that apart from common redox targets such as Glrx1, key proteins in which redox-sensitive thiols were significantly altered are mitochondrial Ldhb and energy metabolism enzymes Mdh2, Tpi1, and Eno2 (Table 2). Furthermore, we found that the reversibly oxidized thiols of Hpx and Kng1were significantly altered both in Nrf2 and Keap1 knockout mice.

#### 3.2.2. Keap1-KO/WT Reversibly Oxidized Redox Peptides

A similar group of redox-sensitive peptides was significantly altered in Keap1-deficient muscle. The deletion of Keap influences the redox states of mitochondrial proteins (Mdh2, Cs, Ndfuv1), proteins associated with energy metabolism (Ladh, and Nadh). Interestingly, apart from the known proteins that are regulated in response to redox homeostasis, our study indicates that the presence of redox switches in structural proteins such as biglycan and in extracellular proteins such as cathepsin and saposin (see Table 3).

### 3.3. Redox Regulation Is Independent of Global Expression Change

Quantitative analysis suggests that the majority of the redox-sensitive peptides were downregulated (~74%) in Nrf2/Keap1 deletion knockouts. We found six significantly altered proteins common to both Nrf2-KO and Keap1-KO muscles (Table 4). We further investigated if the redox differential expression changes were correlated with the global expression profile of the Nrf2/Keap1-KO system. We found that there were five proteins in Nrf2-WT/KO (Ldhb, Glrx, Tpi1, Kng1, and Mdh2) and six proteins in Keap1-WT/KO (Ldhb, Glrx, Tpi1, Cs, Hpx, and Pbxip1) which were detected and quantified in both global and redox proteomic analysis (Table 5). Interestingly, there was no significant change in the global expression profile of proteins with the corresponding redox peptides that were significantly altered in Nrf2 and Keap1 deletion knockouts, indicating the independent regulation of global expression changes and redox homeostasis.

In their oximouse study, Xiao et al. reported that redox-regulated cysteines exist in local environments that tune the target thiol sidechain for oxidative modification [10]. Since most of the significantly quantified redox cysteines were common in Nrf2 and Keap1 knockouts, we examined if these cysteine sites were selectively modified. Structural comparison analysis revealed that most of the thiol side chains were located in the helical region of protein structure (Figure 2A). We further investigated the presence of common sequence motifs in redox-sensitive peptides. Sequence logo analysis highlighted the presence of charged amino acids lysine, arginine, and aspartate in the proximity of redox-regulated cysteine, which is suggestive of a local electrostatic effect favoring the selection of target cysteine (Figure 2B), as proposed in the previous study by Xiao et. al.

### 3.4. Bioinformatics Analyses

#### 3.4.1. Canonical Pathways

The differential protein expression profile findings are indicative of a distinct regulatory role of Nrf2-deficient muscle compared to Keap1-deficient muscles. We subjected the global expression profile data and redox expression data to canonical pathway analysis to gain insight into the regulatory role of Nrf2-KO and Keap1-KO on the intracellular pathways. We observed that Keap1 deletion showed no significant pathway alteration (Figure 3A). However, Nrf2 deletion elicited several important pathways (Figure 3A), including the critical Nrf2-mediated oxidative stress response pathway that is mediated by the upregulation of four proteins: Vcp, Stip, Nqo, and Afar, and the downregulation of six proteins, including Ras, Usp14, Hsp22/40/90, Gst, Cat, and Txn (Figure 4B). Furthermore, pathway enrichment analysis of Nrf2-KO-deficient mice reveals that among the several quantified proteins, seven crucial proteins of the sirtuin signaling pathway were significantly altered. The proteins that were upregulated were mPTP, an inhibitor of apoptosis, Sdha, and Pfk1 tumor growth. The proteins that were downregulated were the Cpt1 protein for fatty acid oxidation, α-tubulin, and Pgam, which is implicated in NADPH production during ROS accumulation (Figure 3B).

Moreover, several proteins associated with mitochondrial dysfunction were found to be altered. We observed that deletion of Nrf2 resulted in the downregulation of several proteins, including the outer mitochondrial membrane protein Cpt1; Cytoplasmic Aconitase and Cat; proteins from complex I (NADH dehydrogenase), including Ndufa6, Ndub6, Ndufb7, Ndufa12, Ndufs7, Ndufs4, Ndufb8, and Ndufb9 as well as the downregulation of the mitochondrial Fis1 protein that is implicated in Alzheimer’s disease. Additionally, the Complex II proteins Sdha and Sdhd, the complex Atp5f1 protein, and the Complex I (NADH Dehydrogenase) protein Ndua10 were significantly upregulated in the Nrf2 deletion knockouts. The effects of Nrf2 deletion were also manifested in the oxidative phosphorylation pathway, where the majority of the proteins were upregulated. We observed the upregulation of specific proteins in NADH Complex 1 (Nd1, Ndufs2, and Ndufa2), FADH Complex II (Sdha and Sdhd), and Complex V (Atp5f). The only protein that was found to be downregulated was the NADH Complex I protein Ndufa12. The other altered pathways in the Nrf2-KO mouse muscle are energy metabolism pathways, estrogen receptory signaling, the BAG2 signaling pathway, xenobiotic metabolism signaling, the protein ubiquitination pathway, the ARE-mediated mRNA degradation pathway, mTOR signaling, the necroptosis signaling pathway, germ cell-–ertoli cell junction signaling, glucocorticoid receptor signaling, polyamine regulation, and P70S6K signaling (Figure 3A).

#### 3.4.2. Networks Analyses

The proteins differentially expressed in Nrf2-deficient skeletal muscles were further investigated to establish the relationship to disease and function. We employed IPA to study these connections and mapped the altered proteins to cellular dysfunction and myopathies (Figure 5). Most of the proteins that were significantly altered were associated with motor dysfunction or movement disorders (indicating the important up/downregulated proteins). Similarly, several proteins that were upregulated and downregulated were implicated in mitochondrial disorders and respiratory chain deficiencies. Finally, seven proteins were associated with metabolic myopathy where Cacna1s, Ampd1, Pygm, Gys1, and Pfkm were upregulated and where Gbe1 and Ca2 were downregulated.

## 4. Discussion

SkM is a highly heterogeneous tissue comprised of fibers with different morphological, functional, and metabolic properties [11]. Mouse SkM contains up to four myosin isoforms, the Type I, IIa, IIx, and IIb, which characterize four pure fiber types (types I, IIA, IIX, and IIB) and at least three hybrid fibers (types I/IIA, IIA/IIX, and IIX/IIB) [12]. These fibers constitute a complete spectrum of contractility from the slowest twitch Type I fibers to the fastest Type IIB fibers, with hybrid fibers being intermediate [13]. Matched with these functional properties, SkM fibers also vary in energy production, with the Type I and IIA fibers primarily using oxidative metabolism and Type IIX and IIB fibers primarily relying upon glycolytic metabolism. On the other hand, multiple fiber types are generally intermingled within a single muscle group, and different muscle groups have varying proportions of fiber types. For example, Sol is a predominantly type IIA fiber, whereas the EDL is predominantly type IIB. These proportions are plastic; however, they occur in response to different physiological and pathological conditions. Endurance exercise training can induce a modest increase in the proportion of Type I fibers contributing to SkM beneficial adaptation, whereas chronic heart failure and aging cause a switch of oxidative fibers to the glycolytic underlying the exercise intolerance of this syndrome. The diversity in contractility and metabolism of different fiber types along with fiber-type plasticity not only provide a wide range of physiological functions in different SkM groups but also result in the differential susceptibility of the SkM group to pathological conditions.

On the other hand, SkM is also highly dynamic both functionally and metabolically. It exhibits a remarkable ability to change its metabolic rate to meet the functional demands imposed on it. Under resting conditions, SkM requires a relatively low energy supply, which can be quickly increased by 100-fold during periods of intense contraction. Indeed, in the course of intense contractile activity, such as during strenuous exercise, oxygen consumption by SkM is extremely high, leading to robust ROS generation, which can result in potential myocyte dysfunction. To counteract this challenge, SkM contains a powerful antioxidant network consisting of enzymatic and non-enzymatic antioxidants, most of which are known to be induced by activating Nrf2. In the hind limb muscles of C57BL/6J mice, 6 hours of treadmill exercise led to the release of Nrf2 from Keap1 to translocate into the nucleus and upregulated the mRNA expression of SOD1, SOD2, Cat, HO-1, GCLc, and GCLm [14]. In a cultured SkM cell line, C2C12, electrical stimulation evoked Nrf2 activation, and the upregulation of NQO1, HO-1, and GCLm mRNAs were significantly impaired when Nrf2 was knocked down by siRNA [15].

Employing mass spectrometry to analyze Nrf2- or Keap1-deficient SkM of iMS-Nrf2flox/flox or iMS-Keap1flox/flox mouse lines, we previously found that Nrf2 knockout altered protein expression in EDL, a fast-twitch muscle with predominantly glycolytic fibers (56% IIB, 24% IIX, and 20% others). In contrast, Keap1 knockout-induced protein expressionmostly appeared in Sol, a slow-twitch muscle with predominantly oxidative fibers (49% IIA, 31% I, and 20% others) [4]. In the present study, we further demonstrated that the RTA, another fast-twitch muscle with predominantly glycolytic fibers (45% IIX, 25% IIB, and 20% others), also displayed a remarkably altered protein expression in response to Nrf2 deletion, which was also observed for the EDL, confirming a higher basal activity of Nrf2 signaling in the glycolytic muscle groups (EDL and RTA) compared to in an oxidative muscle (Sol). Indeed, we have reported that compared to Sol, EDL expresses a higher level of Nrf2 protein and displays a higher response to the Nrf2 activator, the curcumin [16].

Canonical pathway analysis of these identified proteins in Nrf2-deficient RTA suggests that eleven intracellular signaling pathways are altered (Figure 3). As expected, Nrf2 deletion causes significant inhibition of the Nrf2-mediated oxidative stress response due to the downregulation of CAT, TXN, GST, HSP22/40/90, and USP14 proteins. This change can explain the poor muscle contractility and exercise performance observed in this mouse model [4]. Other intracellular signaling pathways suppressed by Nrf2 knockout include oxidative phosphorylation, the BAG2 signaling pathway, the inhibition of the ARE-mediated mRNA degradation pathway, mTOR signaling, and p70S6K signaling. ATP is required for multiple physiological functions of SkM, such as membrane excitability by Na+/K+ ATPase, sarcoplasmic reticulum calcium handling by Ca2+ ATPase, and myofilament cross-bridge cycling by myosin ATPase. Due to the relatively small intramuscular stores of ATP, SkM has to resynthesize ATP to ensure rapid ATP provision and the maintenance of the ATP content in muscle cells to match ATP demand during contraction. This process is involved in the activation of several metabolic pathways where oxidative phosphorylation is predominant [17]. Accordingly, Nrf2 deletion-induced inhibition of oxidative phosphorylation can be considered as one of the mechanisms underlying the impaired contractility and exercise performance in SkM Nrf2-deficient mice. The mammalian target of rapamycin (mTOR) is an evolutionarily conserved serine/threonine kinase and serves as a central regulator of cell metabolism, growth, proliferation, survival [18], and mitochondrial function and structure regulation [19]. mTOR is mainly modulated by its upstream regulator, the PI3K/Akt signaling pathway, and its downstream targeting components include ULK1, CLIP-170, 4E-BP1, S6K1, PP2A, Lipin-1, and SREBP1 [18]. In the present study, we found a downregulation of the mTOR and p70S6K signaling pathways in Nrf2-deficient SkM, suggesting a novel mechanism of Nrf2-induced SkM mass maintenance and growth via promoting mTOR/p70S6K-induced protein synthesis [17,20]. Indeed, it was reported that at least in neurons, Nrf2 can bind to the ARE of the mTOR promoter, leading to the upregulation of mTOR expression and activity [21]. Among the five activated intracellular signaling pathways observed in Nrf2-deficient RTA, the sirtuin signaling pathway and the necroptosis signaling pathway are of interest. Sirtuins are a group of seven highly conserved protein deacetylases that are involved in the process of chromatin remodeling and gene regulation [22]. They have also been shown to have pathophysiological relevance in SkM [23], potentially through the negative regulation of IGF-I and associated signaling pathways [24]. More interestingly, it has been proposed that Sirt1 may reduce SkM mass and growth through inhibiting the mTOR signaling pathway [25]. Traditionally, in contrast with apoptosis, necrosis was considered to be an extreme condition-induced passive death of cells that was independent on the intracellular signaling pathways and characterized by increased cell volume, organelle shrinkage, and plasma membrane disintegration. However, a programmed necrosisfrom named necroptosis was identified in 2005 [26] that is regulated by the receptor interacting protein kinase 1 (RIP1), RIP3, and mixed lineage kinase domain-like pseudokinase (MLKL) signaling pathways [27]. It has been recently demonstrated that necroptosis is a mechanism underlying myofibre death in dystrophin-deficient mice [28]. Taking together, these canonical data suggest that Nrf2 deficiency-evoked myopathy can be induced not only by the loss of cytoprotective mechanisms but also by the activated detrimental signaling pathways.

While oxidative stress plays an important role in SkM pathology by oxidatively damaging proteins, lipids, and DNA through irreversibly altering the structure of these macromolecules, reversible redox post-translational modifications (PTMs) by ROS are mostly appreciated as a regulatory mechanism underpinning muscle adaptation to exercise- and myopathy-induced by chronic diseases and aging [29]. Indeed, increasing evidence has suggested that redox PTMs can modify mitochondrial, myofibrillar, and excitation–contraction (EC) coupling proteins, leading to the alteration of biochemical, electrical, and mechanical characteristics of SkM [30]. The mitochondria is considered to be attractive targets of redox PTMs because the mitochondria is not only an important source of superoxide and H_2_O_2_ production but also has an active thioredoxin (Trx)/GSH redox buffering system and constitutes of many cysteine-enriched proteins, such as 2-oxoglutarate dehydrogenase, isocitrate dehydrogenase, Complex I, and glutaredoxin isoforms 1 and 2 [31]. It has been shown that Complex I activity in cardiomyocytes was reversibly inhibited due to increased S-glutathionylation by diamide treatment [32] or by the knocking out of Grx2 [33]. Titin, a giant elastic protein expressed in the contractile units of striated muscle cells, is another interesting target of redox PTMs [34]. Titin has multiple cysteine residues, which are buried on the interior of the protein during muscle contraction. Under stretching, the cysteines of the titin protein are exposed, making them susceptible to redox PTMs [35]. The reversible PTM modification of cysteines can reduce the elasticity of titin, contributing to impaired relaxation and diastolic dysfunction under conditions of chronic oxidative stress such as aging.

In our previous and present studies, we found that SkM Nrf2 knockout or overexpression can change not only the redox status but also the PTMs-associated enzymes, such as glutathione-S-transferase alpha 2 and 4 (Gsta 2 and 4), glutathione S-transferase Mu 1 and 2 (Gstm 1 and 2), protein-systeine N-palmitoyltransferase HHAT-like protein (Hhatl), throredoxin reductase 1 (Txnrd1), and many others. Accordingly, we further investigated the protein cysteine oxidation profiles of SkM with Nrf2 knockout or overexpression and identified several redox PTMs proteins. Interestingly, we found that redox PTM triosephosphate isomerase (Tpil) was increased by Nrf2 knockout but decreased by Nrf2 overexpression, whereas the redox PTM glutaredoxin-1 displayed an opposite alteration, suggesting multiple factors contribute to the protein redox PTM in SkM. However, the physiological and pathological implication of these protein redox PTM remain to be elucidated in the future.

## 5. Conclusions

The present study suggests that SkM Nrf2 not only regulates cytoprotective gene expression but also modulates protein PTMs, both of which may underlay the Nrf2-involved alteration in SkM metabolism, contractility, and structure under both physiological and pathological conditions.

## Figures and Tables

**Figure 1 genes-12-01655-f001:**
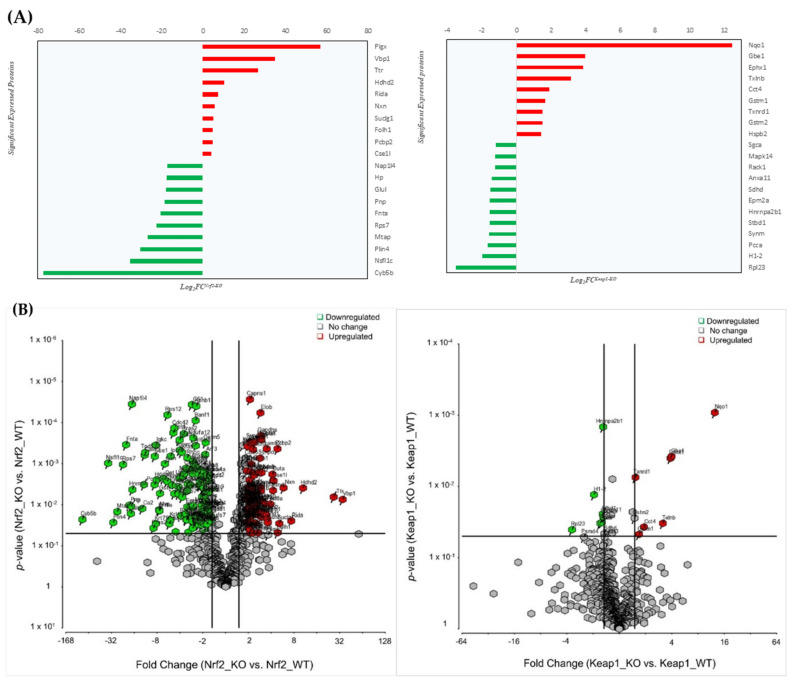
Schemes follow the same formatting. Mass spectrometry-based proteomics analysis of muscles in Nrf2-KO, Nrf2-WT, Keap1-KO, and Keap1-WT mouse models. (**A**) Left panel: top 10 significantly upregulated (red) and downregulated (green) proteins in Nrf2-KO/WT muscles; Right panel: Top 10 significantly upregulated (blue) and downregulated (red) proteins in Keap1-KO/WT muscles. (**B**) Volcano figure plots showing the log2 fold change (WT/KO) plotted against the –log10 *p* value, highlighting significantly changed proteins (red and green; *p* ≤ 0.05 and an absolute fold change of 1.5, *n* = 4, moderated *t*-test). The vertical lines correspond to the absolute fold change of 1.5, and the horizontal line represents a *p* value of 0.05. Log_2_ fold changes in Nrf2-KO and Keap1-KO knockouts are represented as Log_2_FC^Nrf2-KO^ and Log_2_FC^Keap1-KO^, respectively.

**Figure 2 genes-12-01655-f002:**
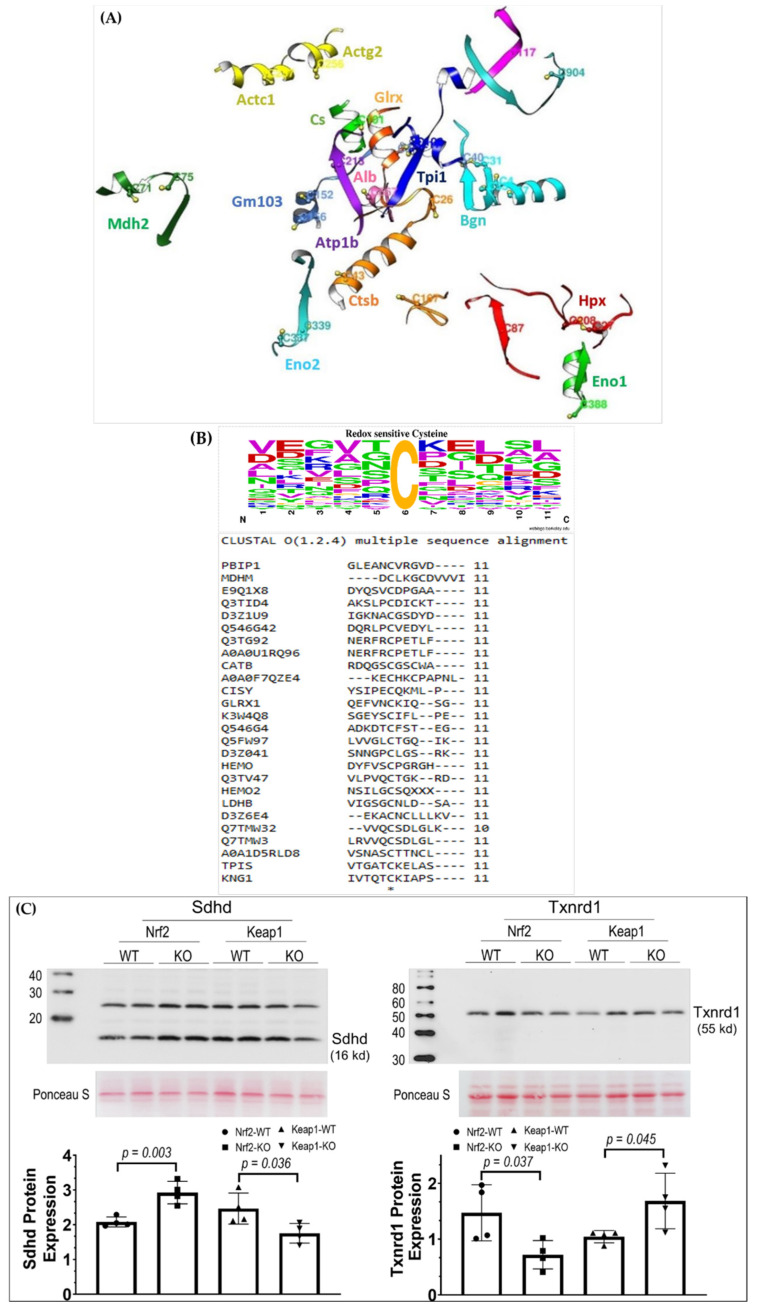
Structural comparison and sequence motif analysis of redox cysteine position in Nrf2 and Keap1 knockouts. (**A**) Superimposition of 3-D structure of peptides harboring redox cysteine (highlighted in ball and stick model). (**B**) Primary sequence motifs for modified cysteine residues. The central ‘‘C’’ represents the redox cysteine residue. The size of the letter represents the probability of the residue. (**C**) Western blot analysis of Sdhd (left panels) and Txnrd1 (right panels) of muscle in Nrf2-WT, Nrf2-KO, Keap1-WT, and Keap1-KO mouse models. Up panels: raw data of Western blots; Bottom panels: group data showing as mean ± SD; *n* = 4 for each group.

**Figure 3 genes-12-01655-f003:**
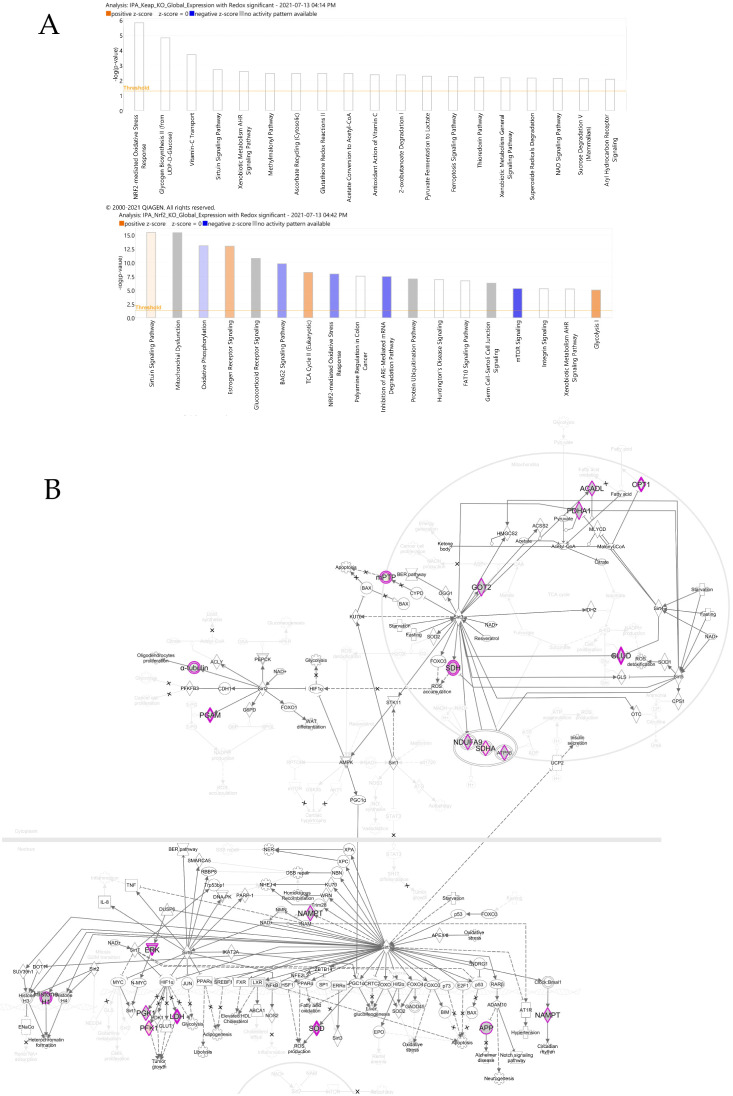
Canonical pathway analysis of differentially expressed proteins in Nrf2-KO and Keap1-KO. (**A**) Benjamini–Hochberg false discovery rate-corrected IPA pathways differentially expressed in Nrf2 KO (upper panel) and Keap1-KO mice (lower panel). The x-axis represents the IPA pathways that were identified, and the y-axis shows the –log *p* of the value calculated based on Fisher’s exact test with multiple correction. Orange bar indicates that the pathway was activated. Blue bar indicates that the pathway was inhibited. Grey bar indicates that there is no measurable change. (**B**) Nrf2-KO proteins significantly differentially expressed in sirtuin signaling pathway. Upregulated/downregulated proteins are highlighted in magenta color.

**Figure 4 genes-12-01655-f004:**
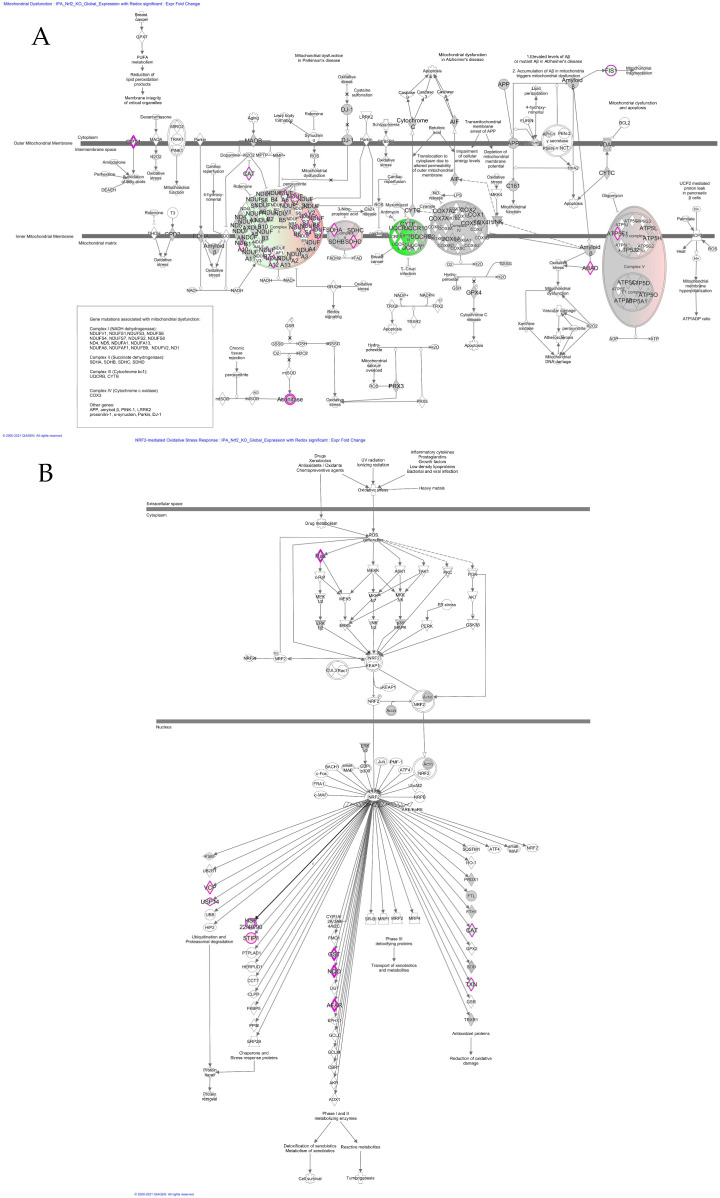
Nrf2-KO proteins significantly differentially expressed in energy metabolism and oxidative phosphorylation. (**A**) Mitochondrial dysfunction and oxidative phosphorylation pathway and (**B**) Nrf2-mediated oxidative stress response pathway. Upregulated/downregulated proteins are highlighted in magenta color.

**Figure 5 genes-12-01655-f005:**
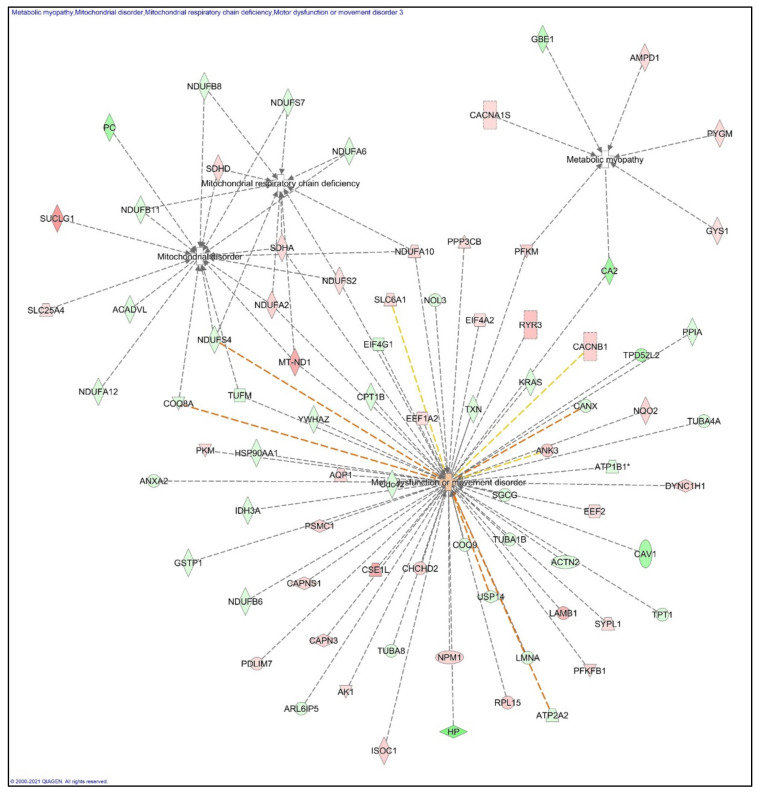
Graphical representation of significantly differentially expressed proteins in Nrf2-deficient skeletal muscles associated with disease and functions. The shapes represent the molecular classes of the proteins (for details visit www.qiagen.com, accessed on 21 June 2019). The color is used to indicate the direction of regulation (red, upregulated; green, downregulated), while the color shade indicates the relative magnitude of change in protein expression levels. The functional interaction networks of the proteins were generated using IPA (QIAGEN Inc., Hilden, Germany).

**Table 1 genes-12-01655-t001:** Common significantly differentially expressed proteins in Nrf2- and Keap1-deficient mice.

Accession	Protein	Log_2_FC^Nrf2-KO^	Log_2_FC^Keap1-KO^
Q9CXV1	Sdhd	1.87569 ^a^	−1.48549 ^b^
Q91ZA3	Pcca	1.69494 ^a^	−1.65623 ^b^
Q70IV5	Synm	1.67506 ^a^	−1.58577 ^b^
Q9JMH6	Txnrd1	−1.38643 ^b^	1.5322 ^a^
P15626	Gstm2	−2.18014 ^b^	1.4955 ^a^
Q9D6Y9	Gbe1	−8.56657 ^b^	3.97791 ^a^

^a^: upregulated; ^b^: downregulated.

**Table 2 genes-12-01655-t002:** List of significantly differentially expressed redox peptides in Nrf2-KO-deficient muscle.

Accession	Protein	Log_2_FC^Nrf2-KO^	Peptide
Q5FW97	EG433182	1.71097 ^a^	[R].SGETEDTFIADLVVGLCTGQIK.[T]
P17751	Tpi1	1.33625 ^a^	[K].VSHALAEGLGVIACIGEK.[L]
D3Z6E4	Eno2	1.29545 ^a^	[K].ACNCLLLK.[V]
P17751	Tpi1	1.20501 ^a^	[R].IIYGGSVTGATCK.[E]
Q546G4	Alb	−1.08798 ^b^	[R].LPCVEDYLSAILNR.[V]
P16125	Ldhb	−1.17278 ^b^	[R].VIGSGCNLDSAR.[F]
Q546G4	Alb	−1.20815 ^b^	[K].AADKDTCFSTEGPNLVTR.[C]
K3W4Q8	Bsg	−1.23564 ^b^	[R].SGEYSCIFLPEPVGR.[S]
Q9QUH0	Glrx	−1.32358 ^b^	(-).MAQEFVNCK.[I]
Q7TMW3	Bgn	−1.36098 ^b^	[R].VVQCSDLGLK.[T]
O08677	Kng1	−1.38058 ^b^	[R].ENEFFIVTQTCK.[I]
P08249	Mdh2	−1.58447 ^b^	[K].GCDVVVIPAGVPR.[K]
Q91X72	Hpx	−1.69061 ^b^	[K].VNSILGCSQ.(-)
Q3TV47	Atp1b1	−1.81841 ^b^	[K].YNPNVLPVQCTGK.[R]
Q91X72	Hpx	−2.73355 ^b^	[R].DYFVSCPGR.[G]

^a^: Upregulated; ^b^: Downregulated.

**Table 3 genes-12-01655-t003:** List of significantly differentially expressed redox peptides in Keap1-KO-deficient muscle.

Accession	Protein	Log_2_FC^Keap1-KO^	Peptide
Q9QUH0	Glrx	2.68731 ^a^	(-).MAQEFVNCK.[I]
Q9CZU6	Cs	−1.34429 ^b^	[R].GYSIPECQK.[LM]
Q3TG92	Actc1	1.99585 ^a^	[R].CPETLFQPSFIGMESAGIHETTYNSIMK.[C]
D3Z1U9	Ndufv1	−1.42557 ^b^	[K].NACGSDYDFDVFVVR.[G]
A0A0F7QZE4	HC	−1.87462 ^b^	[K].CPAPNLEGGPSVFIFPPNIK.[D]
E9Q1X8	Cacna2d1	1.18133 ^a^	[K].SYDYQSVCDPGAAPK.[Q]
Q3TG92	Actc1	1.89048 ^a^	[R].CPETLFQPSFIGMESAGIHETTYNSIMK.[C]
D3Z041	Acsl1	−1.59049 ^b^	[R].GIQVSNNGPCLGSR.[K]
Q3TID4	Psap	−1.39122 ^b^	[K].SLPCDICK.[T]
P17751	Tpi1	−1.79961 ^b^	[R].IIYGGSVTGATCK.[E]
Q91X72	Hpx	−1.44668 ^b^	[K].VNSILGCSQ.(-)
P17751	Tpi1	−1.17728 ^b^	[R].IIYGGSVTGATCK.[E]
P10605	Ctsb	−1.52989 ^b^	[R].DQGSCGSCWAFGAVEAISDR.[T]
A0A1D5RLD8	Gm10358	−1.36198 ^b^	[K].IVSNASCTTNCLAPLAK.[V]
A0A0U1RQ96	Actg2	1.45841 ^a^	[R].CPETLFQPSFI.(-)
P16125	Ldhb	−1.63343 ^b^	[R].VIGSGCNLDSAR.[F]
Q7TMW3	Bgn	−1.23957 ^b^	[R].VVQCSDLGLK.[T]
Q3TVI8	Pbxip1	−1.14557 ^b^	[R].LQGLEANCVR.[G]
Q546G4	Alb	−1.14942 ^b^	[K].AADKDTCFSTEGPNLVTR.[C]

^a^: Upregulated; ^b^: Downregulated.

**Table 4 genes-12-01655-t004:** Common Nrf2/Keap1 knockout redox sensitive significantly differentially expressed peptides.

Accession	Protein	Log_2_FC^Nrf2-KO^	Log_2_FC^Keap1-KO^	Peptides
Q9QUH0	Glrx	−1.32358 ^b^	2.68731 ^a^	(-).MAQEFVNCK.[I]
P17751	Tpi1	1.20501 ^a^	−1.79961 ^b^	[R].IIYGGSVTGATCK.[E]
Q91X72	Hpx	−1.69061 ^b^	−1.44668 ^b^	[K].VNSILGCSQ.(-)
P16125	Ldhb	−1.17278 ^b^	−1.63343 ^b^	[R].VIGSGCNLDSAR.[F]
Q7TMW3	Bgn	−1.36098 ^b^	−1.23957 ^b^	[R].VVQCSDLGLK.[T]
Q546G4	Alb	−1.20815 ^b^	−1.14942 ^b^	[K].AADKDTCFSTEGPNLVTR.[C]

^a^: Upregulated; ^b^: Downregulated.

**Table 5 genes-12-01655-t005:** List of differentially expressed redox peptides in Nrf2/Keap1-KO-deficient muscle and corresponding global protein expression profile.

Expression	Accession	Protein	*p*-Value	Log_2_FC^Nrf2-KO^
Global	P16125	Ldhb	0.389251	−1.43312 ^b^
Redox	P16125	Ldhb	0.0241312 *	−1.17278 ^b^

Global	Q9QUH0	Glrx	0.313294	1.33082 ^a^
Redox	Q9QUH0	Glrx	0.0174908 *	−1.32358 ^b^

Global	P17751	Tpi1	0.350314	−1.2886 ^b^
Redox	P17751	Tpi1	0.0021633 *	1.20501 ^a^

Global	O08677	Kng1	0.0371664 *	−2.70707 ^b^
Redox	O08677	Kng1	0.023165 *	−1.38058 ^b^

Global	P08249	Mdh2	0.249345	1.48018 ^a^
Redox	P08249	Mdh2	0.000200143 *	−1.58447 ^b^

Global	P16125	Ldhb	0.389161	−1.64106 ^b^
Redox	P16125	Ldhb	0.029354 *	−1.63343 ^b^

Global	Q9QUH0	Glrx	0.737846	−1.12477 ^b^
Redox	Q9QUH0	Glrx	2.50 × 10^−5^ *	2.68731 ^a^

Global	P17751	Tpi1	0.817331	1.12616 ^a^
Redox	P17751	Tpi1	0.024665 *	−1.17728 ^b^

Global	Q9CZU6	Cs	0.996303	1.00105 ^a^
Redox	Q9CZU6	Cs	0.000104 *	−1.34429 ^b^

Global	Q91X72	Hpx	0.1457	−1.68804 ^b^
Redox	Q91X72	Hpx	0.011608 *	−1.44668 ^b^

Global	Q3TVI8	Pbxip1	0.25651	−1.80622 ^b^
Redox	Q3TVI8	Pbxip1	0.04031 *	−1.14557 ^b^

* *p* ≤ 0.05; significant change (^a^: Upregulated; ^b^: Downregulated.).

## Data Availability

Not applicable.

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
