# Peer review of "A Quantitative Proteomics Approach to Gain Insight into NRF2-KEAP1 Skeletal Muscle System and Its Cysteine Redox Regulation"

_genes, 2021, doi:10.3390/genes12111655_

Round 1

Reviewer 1 Report

In this manuscript, the authors investigated the global protein expression profiles of Nrf2 and Keap1 deficiency mice. This informative quantitative proteomics results suggested that Nrf2 and Keap1 may be involved in oxidative stress response as well as mitochondrion dysfunction in SkM.

Points:

  1. The authors may want to confirm the sensitivity and accuracy of their methods using western blots on some critical proteins that were identified by Mass spectrometry. For example: the six proteins that were listed in table 1.
  2. Most of these changes listed in tables 2-5 are very subtle, how meaningful are these changes? The authors may also want to confirm some of these results using western blot.

Author Response

We thank the Reviewer for the positive comments and valuable suggestions of our manuscript. We also understand the concerns and critiques of the study that have helped to improve this manuscript. Our responses are indicated below, and the corresponding revisions are marked as Red Text in the manuscript.

  1. The authors may want to confirm the sensitivity and accuracy of their methods using western blots on some critical proteins that were identified by Mass spectrometry. For example: the six proteins that were listed in table 1.

Res: Yes, we agree with the reviewer that the sensitivity and accuracy of mass spectrometry used in this experiment should be validated by different biochemistry methods. Employing western blotting, we did observe a similar change in the protein expression level of Sdhd, and Txnrd1 with the results of mass spectrometry. These new data have been added to the revision shown in Figure 2C.

  1. Most of these changes listed in tables 2-5 are very subtle, how meaningful are these changes? The authors may also want to confirm some of these results using western blot.

Res: The changes in the expression level of all proteins reported in Tables 2 – 5 are statistically significant, although some are small. We agree that, in general, the functional performance of a protein is largely dependent on its absolute amount. These proteins include enzymes involved in the metabolisms of carbohydrates, lipids, and proteins. However, small changes in other proteins triggering intracellular signaling pathways or governing gene expressions may also considerately alter cellular functions through the amplification effects during cascade reactions. The validation of mass spectrometry by western blot has been done in response to comment 1.

Reviewer 2 Report

Abu and colleagues studied the proteomic profiles in the red tibialis anterior muscle (RTA) from mice with conditional skeletal muscle deletion of the Nrf2 and Keap1 genes, and compared them to wild type controls. Nrf2-Keap1 gene products play key roles in the muscle response to oxidative stress, stimulating the expression of cytoprotective downstream genes. In addition, the authors identified proteins with oxidized cysteines as a measure of the changes in redox status following muscle-specific ablation of these two genes.

The levels of almost 300 proteins were altered in Nrf2KO RTA while about 10 times fewer were so in Keap1KO RTA. Pathway analysis showed that these proteins are involved in 4 major signaling cascades: Sirtuin signaling, Nrf2-dependent oxidative response, oxidative phosphorylation and mitochondrial function. Only 6 proteins were found to be regulated by both Nrf2 and Keap1. Further bioinformatics analysis showed that Nrf2-regulated proteins were linked to motor and movement disorders, mitochondrial and respiratory chain deficiencies, and metabolic myopathy. Thirty four proteins in total were identified with oxidized cysteines in Nfr2 and Keap mutant RTA, suggesting that Nrf2/Keap1 signaling not only regulates transcription of cytoprotective genes but also the redox status of some key targets.

The experiments and analyses are rigorous and comprehensive, and the manuscript is very well written. The following are items to address.

  1. It would strengthen the manuscript if at least a few of the most relevant changes in protein expression detected by MS were validated by independent methods such as Western blotting or immunostaining.
  2. As Nrf2-regulated proteins outnumbered Keap1-regulated ones, and Keap1KO muscle showed no signaling pathway alterations, could the authors further comment on the reasons and implications of these findings. Is Keap1 not essential or redundant for the cytoplasm/nuclear shuttling of Nrf2 in RTA, EDL and similar glycolytic muscles? Are there other proteins that play this role here?
  3. The present study seems to be just a confirmatory extension of prior work on EDL and Soleus by the same authors (Gao et al., 2020). It is unclear what is novel about this study in the RTA; further clarification on the novelty of the present study would be important.
  4. Related to #3, have the authors checked experimentally whether the fiber-type composition of the RTA is not altered by Nrf2 or Keap1 deletion? It seems that the assumption it is that is not.
  5. Line 23: define iodoTMT acronym.
  6. Line 312: typo on “energy”.

The text size within Figs 3 and 4 is just too small and some of the diagrams are so busy with the labeling that it hinders interpretation. It would be appreciated if the authors could find a way to simplify some of the network diagrams.

Author Response

We thank the Reviewer for the positive comments and valuable suggestions of our manuscript. We also understand the concerns and critiques of the study that have helped to improve this manuscript. Our responses are indicated below, and the corresponding revisions are marked as Red Text in the manuscript.

Abu and colleagues studied the proteomic profiles in the red tibialis anterior muscle (RTA) from mice with conditional skeletal muscle deletion of the Nrf2 and Keap1 genes, and compared them to wild type controls. Nrf2-Keap1 gene products play key roles in the muscle response to oxidative stress, stimulating the expression of cytoprotective downstream genes. In addition, the authors identified proteins with oxidized cysteines as a measure of the changes in redox status following muscle-specific ablation of these two genes.

The levels of almost 300 proteins were altered in Nrf2KO RTA while about 10 times fewer were so in Keap1KO RTA. Pathway analysis showed that these proteins are involved in 4 major signaling cascades: Sirtuin signaling, Nrf2-dependent oxidative response, oxidative phosphorylation and mitochondrial function. Only 6 proteins were found to be regulated by both Nrf2 and Keap1. Further bioinformatics analysis showed that Nrf2-regulated proteins were linked to motor and movement disorders, mitochondrial and respiratory chain deficiencies, and metabolic myopathy. Thirty four proteins in total were identified with oxidized cysteines in Nrf2 and Keap1 mutant RTA, suggesting that Nrf2/Keap1 signaling not only regulates transcription of cytoprotective genes but also the redox status of some key targets.

The experiments and analyses are rigorous and comprehensive, and the manuscript is very well written.

Res: Thank you for these positive comments.

  1. It would strengthen the manuscript if at least a few of the most relevant changes in protein expression detected by MS were validated by independent methods such as Western blotting or immunostaining.

Res: Thank you, and we agree. Please see our response to comment 1 of Reviewer #1.

  1. As Nrf2-regulated proteins outnumbered Keap1-regulated ones, and Keap1KO muscle showed no signaling pathway alterations, could the authors further comment on the reasons and implications of these findings. Is Keap1 not essential or redundant for the cytoplasm/nuclear shuttling of Nrf2 in RTA, EDL and similar glycolytic muscles? Are there other proteins that play this role here?

Res: Thank you for raising these great questions. Yes, it looks that the RTA displays a similar phenotype with EDL following Nrf2 or Keap1 deletion that is distinct from the Sol. We agree with you that our data do suggest that Keap1 may play a lesser role in the regulation of Nrf2-antioxidant defense in glycolytic muscles than that in oxidative muscles. However, this does not mean that Nrf2 is not important in this type of muscle. On the contrary, Nrf2 is essential for the redox homeostasis of RTA where, we found, Nrf2 deletion altered expression of 298 proteins and activity of 11 intracellular signaling pathways. We completely agree that there should be other mechanisms than Keap1 to modulate Nrf2 in these glycolytic muscles that, if identified, will be significant progress in the field. We will keep our eyes on this point.

  1. The present study seems to be just a confirmatory extension of prior work on EDL and Soleus by the same authors (Gao et al., 2020). It is unclear what is novel about this study in the RTA; further clarification on the novelty of the present study would be important.

Res: Thank you for this comment. Although the animal models and primary technique are the same in these two studies, the innovation of the present study lies in several aspects. As indicated in the manuscript, Sol and EDL are constituted with type IIA and IIB fibers separately, whereas the RTA is constituted predominantly with type IIX fiber. Accordingly, the current study is not a duplication of previous one simply by using a different muscle mass. The data obtained from the present study represents the significance of Nrf2-Keap1 mediated antioxidant defense and other cytoprotective effects in a new fiber type. Indeed, in this study, we identified some new Nrf2 target proteins and intracellular signaling pathways, which did not be reported in our previous study or from other labs. Another innovation of this study is to quantitatively analyze redox-modified proteins. As we know, reversible redox post-translational modifications (PTMs) play a critical role in regulating protein function. In the current study, by utilizing the iodoTMT reagent labeling technique, we identified 15 proteins in Nrf2 deficient muscle and 19 proteins in Keap1 deficient muscle with the reversibly oxidized cysteine residues. These data provide a deeper insight into the Nrf2/Keap1 system's significance in redox biology that was not reported in our previous studies or from other laboratories.   

  1. Related to #3, have the authors checked experimentally whether the fiber-type composition of the RTA is not altered by Nrf2 or Keap1 deletion? It seems that the assumption it is that is not.

Res: We agree that this is a very interesting question. Long-term alterations in physiological or pathological conditions have been demonstrated to lead to changes in fiber-type composition of skeletal muscle. Six weeks of voluntary wheel running induced a significant fiber type IIb to IIa/x shift in triceps muscles of mice (Rockl K, et al. Diabetes 56:2062–2069, 2007), whereas the rats with congestive heart failure displayed a slow-to-fast phenotype transition in Sol (Sousa E, et al. Circulation. 2000;102:1847-1853). Yes, we assumed that this phenomenon will not occur in the conditional gene knockout models used in the present study due to a relatively shorter period of genetic modification induced by tamoxifen administration. However, I think this is a very good point and we will clarify it in the future project.

  1. Line 23: define iodoTMT acronym.

Res: Thank you. We have incorporated the same in the manuscript

  1. Line 312: typo on "energy".

Res: Thank you. The text has been modified suitably.

  1. The text size within Figs 3 and 4 is just too small and some of the diagrams are so busy with the labeling that it hinders interpretation. It would be appreciated if the authors could find a way to simplify some of the network diagrams.

Round 2

Reviewer 2 Report

I thank the authors for providing evidence of independent validation of their MS data using Western Blotting and for their efforts to simplify some of the more complex figures in the manuscript.